# Participation in community-based health care interventions (CBHIs) and its association with hypertension awareness, control and treatment in Indonesia

Sujarwoto Sujarwoto [1]*, Asri Maharani[2,3]

1 Portsmouth Brawijaya Center for Global Health, Population and Policy & Department of Public Administration, Universitas Brawijaya, Malang, Indonesia, 2 Division of Nursing, Midwifery, & Social Work, University of Manchester, Manchester, United Kingdom, 3 Faculty of Medicine, Universitas Brawijaya, Malang, Indonesia

* sujarwoto@ub.ac.id

## Abstract

**Data Availability Statement:** The data underlying the results presented in the study are available from https://www.rand.org/well-being/social-and-behavioral-policy/data/FLS/IFLS/access.html.

### Background

Little attention has been paid to whether CBHIs improve awareness, treatment and control of hypertension in the contexts of low- and middle- income countries (LMICs). This study therefore aims to examine participation in CBHIs for non-communicable diseases (NCDs) and its association with awareness, treatment, and control of hypertension among Indonesians.

### Methods

This study used data from the 2014 Indonesia Family Life Survey (IFLS), drawn from 30,351 respondents aged 18 years and older. Participation in CBHIs was measured by respondents' participation in CBHIs for NCDs (*Posbindu PTM* and *Posbindu Lansia*) during the 12 months prior to the survey. Logistic regressions were used to identify the relationships between participation in CBHIs for NCDs and awareness, treatment, and control of blood pressure among respondents with hypertension.

### Results

The age-adjusted prevalence of hypertension was 31.2% and 29.2% in urban and rural areas, respectively. The overall age-adjusted prevalence was 30.2%. Approximately 41.8% of respondents with hypertension were aware of their condition, and only 6.6% of respondents were receiving treatment. Participation in CBHIs for NCDs was associated with 50% higher odds of being aware and 118% higher odds of receiving treatment among adults with hypertension. There was no significant association between participation in CBHIs for NCDs and controlled hypertension.

**Funding:** SS and AM received funding from Kementerian Riset, Teknologi dan Pendidikan Tinggi, 292.58 UN 10. C10/PN/2020 (The Directorate General of Higher Education, Ministry of Education and Culture, Republic of Indonesia). The funders had no role in study design, data collection and analysis, decision to publish, or preparation of the manuscript.

**Competing interests:** The authors have declared that no competing interests exist.

## Conclusion

Our data emphasise the importance of CBHIs for NCDs to improve the awareness and treatment of hypertension in the Indonesian population.

## Introduction

The presence of NCDs may thwart or delay attainment of the United Nations' Sustainable Development Goals in low- and middle-income countries (LMICs) [1,2]. The 2014 WHO Global Status Report highlighted the alarming increase of NCDs in LMICs, especially cardiovascular diseases, diabetes and cancers [3]. These diseases are the biggest burden on health systems in LMICs, particularly in countries with universal health coverage [3]. In LMICs, one of the major risk factors for NCDs—especially cardiovascular diseases—is hypertension, which is extremely prevalent [4,5]. In 2015, approximately 1.3 billion people worldwide had hypertension; one billion of them resided in LMICs [6,7]. Despite the substantial decrease of mean blood pressure in high-income Western and Asia Pacific countries in the last four decades, blood pressure has increased in across LMICs in East and Southeast Asia. From 2000 to 2010, the age-adjusted prevalence of hypertension decreased by 2.6% in high-income countries but rose by 7.7% in LMICs [7].

As in most other LMICs in the Asia Pacific region, the prevalence of hypertension in Indonesia has been rising. Using the 2007 IFLS, Hussain et al. ascertained that it had reached 47.8% in 2007 among Indonesian adults aged 40 years and older [8]. A separate study focusing on women in urban Indonesia found that, in major Indonesian cities, 31% of women aged 15 years and older had hypertension [9]. Hypertension is a major risk factor for cardiovascular diseases in Indonesia, accounting for 20%-25% of all coronary diseases and 36%-42% of all strokes [10]. Studies have reported that hypertension rates are higher in urban areas than in rural areas as urban residents are more likely to lead sedentary lifestyles, consume unhealthy foods, and smoke [11,12].

The decrease in hypertension rates in high-income countries shows that hypertension can be prevented and controlled through a combination of behavioural, lifestyle, and drug treatment strategies [5]. Hence, increasing awareness, treatment and control of hypertension in LMICs is vital for reducing their high hypertension rates [13,14]. Otherwise, lack of awareness, treatment and control of hypertension can lead to life-threatening complications, impacting health and economic gains [14]. However, an unmet need for treatment and control of hypertension is common in LMICs due to lack of health care manpower, which thus reduces the scope of task sharing [15,16]. Accordingly, CBHIs are often recognised as a unique mode of health care delivery and considered a fundamental element of health care task sharing in LMICs [16]. CBHIs are also seen as an instrument in the creation of healthy community environments through broad systemic changes in public policy and community-wide institutions and services in LMICs [17,18]. Studies have thoroughly documented the roles of CBHIs in health care and health in LMIC contexts, but most have focused on communicable diseases, nutrition and family planning [18,19]. All have documented the benefits of CBHIs in spreading information related to communicable disease prevention as well as access to nutrition and contraception services [19–22].

Most existing studies on the association of CBHIs and NCDs, particularly hypertension awareness, treatment and control, have been conducted in the contexts of high-income countries. Zhang et al. performed a systematic review of 34 studies (24 of them conducted in the US, the UK, Japan, South Korea and Canada) on community hypertension interventions and

found that community programs benefit hypertension awareness, treatment and control via lifestyle modification and medication adherence in most of those high-income countries [23]. Further, scholars have noted two advantages of CBHIs with regard to hypertension awareness, treatment and control: (1) they are able to provide culturally sensitive health education strategies and (2) they are able to deliver essential medical services to patients in the lower level of public health care [23,24].

In LMICs, studies on the association of CBHIs on hypertension awareness, treatment and control show mixed results. Devkota et al. found no association between community-based programs and treatment and control of hypertension in Nepal [25]. A large proportion of the hypertensive population in Nepal remains untreated; the authors found that hypertension control was, however, significantly associated with access to and use of combination therapy, adherence to medication, follow-up care, and availability of health counselling. Concerns about a lifelong need for medication, perceived side effects of drugs, non-adherence to medication, lack of follow-up, and inadequate counselling from physicians were the most common barriers to hypertension control [25]. Based on cross-sectional data from the 2015 China Health and Retirement Longitudinal Study (CHARLS), Song et al. reported that community based blood pressure monitoring services had no association with blood pressure control [26]. In contrast with these findings, Premkumar et al. found a significant association between CBHIs and control of hypertension in India [27]. The authors suggested that the CBHI approach allowed older people to become conscious of the risk of hypertension through health education and encouraged them to comply with their medical regimes, a government goal reinforced by community approaches and structures [27]. Gonzales et al. found a positive association between CBHIs and hypertension treatment and control in the Cuban population. One of Cuba's key success factors is the readiness of health system interventions (i.e. the programs are well equipped and the non-health workers are well trained) as suggested by the WHO HEARTS program [25]. The mixed findings detailed above suggest the need for a better understanding of the association of CBHIs with hypertension awareness, treatment and control in various LMIC contexts.

The case of CBHIs in Indonesia may be a unique one. Indonesia's health care system has been characterised by a broad decentralization that has given local community and government the authority and responsibility to deliver health care [28,29]. One example of this decentralization as related to health care is the establishment of various CBHIs [29,30]. Such community involvement has been the key to success in various government-mandated programs, with village health posts or *Posyandu* representing notable achievements in family planning [29,30]. Following the success of the *Posyandu* program for family planning, the Indonesian government established two CBHIs for NCDs in 2011: integrated health posts for NCDs (*Posbindu PTM*) and integrated health posts for older people (*Posyandu Lansia*) [31]. *Posbindu PTM* were developed specifically for NCD monitoring and counselling in communities; they target the population aged 15 years and older, while *Posyandu Lansia* focus on the health of the elderly (65 years and older) [32]. *Posbindu PTM* personnel consist of 5–8 *Kaders* (trained health volunteers from the local community) technically supervised by local health offices. The five principal activities of the *Posbindu PTM* and *Posbindu Lansia* are (1) anthropometry, (2) blood pressure measurements, (3) blood glucose and cholesterol testing, (4) health counselling and education, and (5) promotion of physical activity and exercise. Prior to beginning their jobs, *Kaders* must attend a three-day training conducted by trained nurses and physicians appointed by the Ministry of Health. The *Posbindu* book of technical guidelines, which consists of a curriculum and materials for training, is provided by the Ministry of Health. The curriculum includes: an introduction to *Posbindu* management, *Kaders'* role and health promotion skills, anthropometry, blood pressure measurement, and blood glucose and

cholesterol measurement [32]. *Kaders* must score >80% on their training evaluation tests in order to be selected as *Posbindu Kaders* [32]. Modest funding of approximately IDR 250K-500K (USD 18–36) is provided by district governments via Primary Health Care Centres (*Puskesmas*) and community village funds. The funds are used to defray the costs of *Kaders'* transportation during the conducting of health promotions and screenings [32]. According to a Ministry of Health report, there were 33,679 *Posbindu PTM* in Indonesia in 2018 [33]. The government target for 2021 is 136,000 CBHIs for NCDs. Early detection of NCDs in Indonesia in 2018 was at approximately 42% of the target population, against the government target of 100% [33]. Since the inception of the program in 2011, few analyses of the potential benefits of CBHIs in hypertension awareness, treatment and control in Indonesia have been undertaken. This study thus aims to examine the association between participation in CBHIs for NCDs and hypertension awareness, treatment and control among Indonesians.

## Materials and methods

### Data

This study used cross-sectional data from the 2014 Indonesia Family Life Survey (IFLS). The survey was representative of approximately 83% of the Indonesian population living in 13 of the country's 26 provinces [34]. The selection of provinces was intended to maximise representation of the population and to capture the cultural and socio-economic diversity of Indonesia [34]. Within each of the 13 provinces, 321 enumeration areas were randomly chosen from a nationally representative sample frame used in the 1993 National Social Economic Survey (*Susenas*), a national representative socio-economic survey of about 250,000 households and 1.25 million individuals [34]. The IFLS then over-sampled urban enumerator areas and enumerator areas in smaller provinces to facilitate urban–rural and Javanese–non-Javanese comparisons [34]. In total, the 2014 IFLS collected information from 50,144 individuals (51% women) of all ages (0 to 80+) from 16,931 households [34]. According to the follow-up report, "the dynasty re-contact rate was 92%. For the individual target households (including split off households as separate) the re-contact rate was [. . .] 90.5%" [34]. For the purposes of our analysis, information from 30,051 respondents aged 18 years and older was included [34]. The survey protocol was reviewed and approved by the Institute Review Board at RAND in the United States and by the University of Gadjah Mada in Indonesia [34]. Written informed consent was obtained from all respondents before data collection. Prior to interviewing, respondents were informed as to why it was important for them to participate in the study and were provided with "examples of policies that have been affected by the study" [35]. Confidentiality and anonymity were ensured.

The survey was designed between October 2012 and April 2014. The pretest of the questionnaire was conducted in Central Java from October to November 2013, with Solo representing urban areas and nearby Sukoharjo representing rural areas. The pretest stratified respondents by age, gender and education and focused on questionnaire content, field editing protocols, use of computer-assisted personal interviewing (CAPI) and general field procedures [34]. Its primary objectives were to fully test the questionnaires in the field and separately for an urban and a rural area, to evaluate the length of the questionnaires, and to test the use of the CAPI program for data collection. The CAPI program CSPro was chosen as it is user friendly and allows interviewers to easily navigate between questions as well as to go back in the program to correct any errors. The pretest was conducted by 15 staff members, many of whom had previously served as senior field staff. The PI and co-PI also participated in the pretest. Overall, the results showed that even those with no schooling were able to correctly understand all questions in the questionnaires [34].

All enumerators and staff members underwent a thorough three-week training in the use of the questionnaires. The training for interviewers was conducted in early August 2014. A total of 23 teams were sent into the field; 210 trained enumerators were employed for data collection. An additional 12 staff members worked facilitating the logging in and cataloging of data, coordinating logistics and checking the quality of filled questionnaires. The fieldwork for the data collected was conducted between September 2014 and April 2015 [34].

## Measures of awareness, treatment and control of hypertension

During data collection, respondents' blood pressure was measured three times by specially trained interviewers using OMRON HEM-7130 self-inflating sphygmomanometers with digital read-out. The devices were manufactured in Kyoto, Japan. The OMROM HEM-7130 was last calibrated in August 2015. Normal sized cuffs were used in most cases, while large cuffs were available when needed [34]. The first blood pressure measurement was conducted at the outset of each research interview, with two subsequent assessments taking place during the course of the interview. For each of the three measurements, the average blood pressure was calculated arithmetically from the systolic and diastolic blood pressure [34]. In this study, hypertension was defined by means of a mean systolic blood pressure (SBP) of $\geq$ 140 mmHg and/or diastolic blood pressure (DBP) of $\geq$ 90 mmHg and/or being in antihypertensive drug therapy [36]. Respondents with hypertension were considered to be aware of their condition if they answered "yes" to the interview question: "Has a doctor/paramedic/nurse/midwife ever told you that you had hypertension?" These respondents were considered to be on treatment if they answered "yes" to the interview question: "In order to manage your hypertension, are you currently taking prescribed medication on a weekly basis?" Respondents were considered to have their hypertension under control if they reported that they were on antihypertensive medication and had a mean SBP of $<$ 140 mmHg and DBP of $<$ 90 mmHg [36].

## Participation in CBHIs for NCDs and control variables

We categorised respondents as participating in a CBHI for NCDs if they answered "yes" to the interview question: "During the last 12 months, did you participate in or use *Posbindu PTM* or *Posbindu Lansia* in your community?" A number of sociodemographic characteristics that may covary with participation in CBHIs and awareness, treatment and control of hypertension were included as control variables [37,38]. Age was included to control for whether CBHI participation and awareness, treatment and control of hypertension may vary across age. Age was divided into three categories that capture age as a cardiovascular risk factor: young adult (18–39 years), middle-aged adult (40–59 years) and older adult ($\geq$60 years) [39]. We classified ethnicity as Javanese or non-Javanese (reference group) to control for whether participation in CBHIs varies between Javanese and non-Javanese [40]. Marital status was divided into three groups: single (reference group), married and divorced/widowed [40]. We included education to control whether CBHI participation and awareness, treatment and control of hypertension may vary across respondents' educational levels. Following the Indonesian national education system, educational attainment was divided into three groups: primary school or less (reference group), secondary school, and college or higher [41]. To capture whether economic status may link with awareness, treatment and control of hypertension, we included quintiles of monthly per capita expenditure [42]. We chose to use monthly per capita expenditure rather than income to determine wealth quintile as this measure more accurately captures levels of long-term economic resources [43]. A dummy variable for health insurance was created to examine whether respondents covered by health insurance had more access to hypertension treatment and control than those who were not covered by health insurance [44]. We

categorised respondents as having health insurance if they answered "yes" to the interview question: "Are you currently the beneficiary of any type of health insurance?" Lastly, we included geographical area to determine whether CBHI participation as well as awareness, treatment, and control for hypertension may vary across islands [45]. We categorised geographical areas as follows: Java (reference group), Sumatera, Kalimantan, Sulawesi, and other islands [45].

## Statistical analysis

The statistical analysis was conducted in the following steps: Firstly, descriptive statistics were used to describe the study variables in the study population. Secondly, prevalence analysis was conducted to describe hypertension rates based on residential area as it has been suggested that people in urban areas are at higher risk of leading sedentary lifestyles, consuming unhealthy foods, and smoking [11,12]. Moreover, rural areas in Indonesia are often characterised by a relative lack of access to various public services as compared to urban areas [46,47]. IFLS 2014 used the definition of urban and areas from Indonesia's Spatial Planning Act 2007. An urban area is thus defined as "a place that has major non-agricultural activity and functions as an urban settlement with concentration and distribution of government services, social services and economic activities." In terms of population density, a place is considered urban if it has a minimum population of 5,000, a minimum density of 400 persons per square kilometre (1,000/sq mi), and a minimum of 75% of the male working population employed in non-agricultural activities. A rural area is defined as "an area that has major agricultural activity, including the management of natural resources in the region" [48]. To capture differences across socio-demographic characteristics, we also calculated the prevalence of hypertension in the urban and rural population by education, marital status, and wealth. The data in urban and rural areas were compared using Kruskal-Wallis one-way analysis of variance for numerical variables and ordinal chi-square tests for categorical variables to analyse the differences between residential areas [49]. Survey weight was used in all analyses to adjust for non-response bias. We used the IFLS 2014 cross-sectional analysis person weights for the sampling procedures (which oversampled urban areas and some outer provinces) and for attrition [34]. The prevalence and awareness, treatment and control rates of hypertension were age-adjusted by direct standardisation to the 2010 Indonesia population census. Thirdly, logistic regression analyses were applied to identify the determinants of awareness, treatment and controlled blood pressure among respondents with hypertension. For all statistical analyses, a two-sided $p$-value of $<0.05$ and odds ratios based on 99% confidence intervals were conducted. The maximum likelihood (ML) estimator was used to estimate all models; for the probability model we also reported the odds ratio (OR) [50]. Poisson regressions were conducted for sensitivity analyses since the outcome variable in this study is common. As suggested by Barros and Hirakata, the use of odds ratios when the outcome variable is common may overestimate the measure of association. Thus, models that directly estimate the prevalence ratio are needed [51]. We used the *svy* command in STATA to include the sampling weights in the analysis. STATA 16 was used to estimate the models.

## Results

### Sample characteristics

Table 1 illustrates the descriptive statistics of the 30,351 respondents and compares their characteristics according to residential area. A total of 12,582 and 17,869 respondents resided in urban and rural areas, respectively. The mean age of the respondents was 44.98±15.98 years. More than half of the respondents were female (54.4%), Javanese (53.8%), and/or living on

**Table 1. Socio-demographic characteristics of respondents in rural and urban areas, 2014–2015.**

| Variables, mean or % (standard deviation) | Total (n = 30,351) | Rural (n = 12,582) | Urban (n = 17,869) |
|---|---|---|---|
| Age in years | 44.98 (15.98) | 46.02 (16.10) | 43.91 (15.78) ‡ |
| *Age group, in years* | | | |
| 18–39 | 38.4 (0.3) | 35.6 (0.4) | 41.2 (0.4) ‡ |
| 40–59 | 42.7 (0.3) | 43.8 (0.5) | 41.5 (0.4) |
| ≥60 | 18.8 (0.3) | 20.4 (0.4) | 17.2 (0.3) ‡ |
| Women, % | 54.4 (0.3) | 55.1 (0.5) | 53.8 (0.4) |
| Javanese, % | 53.8 (0.3) | 54.6 (0.5) | 52.9 (0.4) |
| *Educational level, %* | | | |
| Primary school or less | 43.8 (0.3) | 55.7 (0.5) | 32.6 (0.4) ‡ |
| Secondary school | 44.1 (0.3) | 37.4 (0.5) | 50.4 (0.4) ‡ |
| College or higher | 12.0 (0.2) | 6.8 (0.2) | 16.8 (0.3) ‡ |
| *Marital status, %* | | | |
| Single | 10.5 (0.1) | 7.5 (0.2) | 13.6 (0.2) ‡ |
| Married | 75.7 (0.3) | 78.2 (0.4) | 73.2 (0.3) ‡ |
| Divorced/widowed | 13.6 (0.2) | 14.1 (0.3) | 13.1 (0.4) |
| *Wealth quintiles, %* | | | |
| Poorest | 22.2 (0.3) | 27.4 (0.5) | 16.7 (0.3) ‡ |
| Second | 20.9 (0.3) | 24.2 (0.4) | 17.5 (0.3) ‡ |
| Third | 19.4 (0.2) | 19.2 (0.4) | 19.5 (0.3) |
| Fourth | 19.2 (0.2) | 17.5 (0.4) | 21.0 (0.3) ‡ |
| Wealthiest quintile (5th) | 18.1 (0.2) | 11.4 (0.3) | 25.1 (0.3) ‡ |
| *Blood pressure, mmHg* | | | |
| *Systolic* | | | |
| Overall | 132.51 (23.00) | 133.00 (22.92) | 132.02 (23.09) ‡ |
| 18–39 | 120.78 (13.19) | 121.17 (12.97) | 120.43 (13.37) ‡ |
| 40–59 | 135.33 (22.54) | 134.89 (22.02) | 135.84 (23.11) |
| ≥60 | 150.23 (26.35) | 149.82 (26.63) | 150.73 (26.01) |
| *Diastolic* | | | |
| Overall | 80.27 (12.18) | 80.03 (12.04) | 80.52 (12.33) * |
| 18–39 | 76.18 (9.86) | 75.95 (9.61) | 76.38 (10.08) |
| 40–59 | 83.11 (12.63) | 82.63 (12.34) | 83.63 (19.23) † |
| ≥60 | 82.22 (13.10) | 81.62 (13.27) | 77.37 (11.73) * |
| Participation in CBHIs for NCDs, % | 4.2 (0.1) | 3.46 (0.21) | 5.13 (2.19) ‡ |
| Health insurance, % | 45.8 (0.3) | 39.03 (0.52) | 54.26 (0.43) ‡ |
| *Geographical areas, %* | | | |
| Java and Bali | 75.3 (0.2) | 70.0 (0.4) | 80.7 (0.2) ‡ |
| Sumatera | 15.5 (0.2) | 18.8 (0.3) | 12.2 (0.2) ‡ |
| Kalimantan | 3.02 (0.08) | 3.9 (0.1) | 2.0 (0.08) ‡ |
| Sulawesi | 3.2 (0.09) | 4.0 (0.01) | 2.4 (0.09) ‡ |
| Other islands | 2.7 (0.06) | 3.0 (0.1) | 2.5 (0.08) ‡ |

**Notes:** CBHIs = community-based health interventions; NCDs = non-communicable diseases. Data are weighted using survey weight and expressed as percentages (standard error) or means (standard error).

*$p<0.05$

†$p<0.01$

‡$p<0.001$ for the difference between urban and rural areas.

Java or Bali Island (75.3%), and most were married (75.7%). Respondents living in rural areas tended to be older, less educated, and poorer. The means of the systolic and diastolic blood pressure of all respondents were 132.51±23 mmHg and 80.27±12.18 mmHg, respectively. The average systolic blood pressure of respondents in rural areas (133±22.92 mmHg) was slightly higher than those in urban areas (132.02±23.09 mmHg), and the difference was statistically significant. The diastolic blood pressure levels of respondents in rural areas (80.03±12.04 mmHg) were lower than those of respondents in urban areas (80.52±12.33 mmHg), but the difference was not statistically significant. Approximately 4.2% of the respondents had participated in CBHIs for NCDs in the 12 months previous to the interview, and the proportion was higher among those living in urban areas. The proportion of respondents who had health insurance was higher among respondents living in urban (54.26%) than in rural areas (39.03%).

### Prevalence of hypertension

Table 2 shows the prevalence of hypertension among the rural and urban populations by education, marital status, and wealth. The age-adjusted prevalence of hypertension was 30.2%. The age-adjusted prevalence of hypertension in urban areas (31.2%) was higher than in rural areas (29.2%; $p<0.0001$). The prevalence of hypertension increased linearly with age overall, and the increase was more marked among those living in urban areas. The prevalence of hypertension was higher among males in urban areas than in rural areas ($p<0.001$), but this difference did not appear among females. Not having attended school or having obtained only

Table 2. Prevalence of hypertension in the Indonesian population, 2014–2015.

| | Total | | Rural | | Urban | |
|---|---|---|---|---|---|---|
| | Unadjusted | Age- adjusted[a] | Unadjusted | Age- adjusted[a] | Unadjusted | Age- adjusted[a] |
| Overall | 38.3 (0.3) | 30.2 (0.2) | 37.1 (0.5) | 29.2 (0.4) | 37.6 (0.4) | 31.2 (0.3)‡ |
| *Age, in years* | | | | | | |
| 18–39 | 15.7 (0.3) | 15.4 (0.3) | 15.4 (0.5) | 15.1 (0.4) | 15.9 (0.4) | 15.8 (0.3) |
| 40–59 | 43.7 (0.5) | 42.4 (0.5) | 41.8 (0.8) | 40.8 (0.8) | 45.8 (0.7)‡ | 44.0 (0.6)† |
| ≥60 | 67.0 (0.8) | 67.4 (0.8) | 65.0 (1.3) | 65.0 (1.3) | 69.4 (1.1)† | 70.0 (1.1)† |
| *Sex* | | | | | | |
| Male | 33.1 (0.4) | 28.5 (0.3) | 32.2 (0.7) | 26.6 (0.6) | 34.0 (0.6) | 30.4 (0.5)‡ |
| Female | 40.8 (0.4) | 31.1 (0.3) | 41.1 (0.7) | 30.8 (0.5) | 40.6 (0.6) | 31.3 (0.4) |
| *Educational level* | | | | | | |
| Primary school or less | 45.6 (0.6) | 31.5 (0.5) | 43.2 (0.8) | 31.0 (0.7) | 49.3 (0.8)‡ | 32.1 (0.7) |
| Secondary school | 27.3 (0.4) | 29.5 (0.4) | 23.3 (0.7) | 26.9 (0.8) | 30.2 (0.5)‡ | 30.5 (0.5)‡ |
| College or higher | 27.9 (0.8) | 29.7 (0.7) | 27.1 (1.8) | 26.2 (1.4) | 28.1 (0.9) | 30.3 (0.8)* |
| *Wealth quintiles* | | | | | | |
| Poorest | 38.7 (0.8) | 29.3 (0.6) | 36.9 (1.0) | 28.0 (0.8) | 41.6 (1.1)‡ | 31.6 (0.9)† |
| Second | 36.1 (0.8) | 28.9 (0.6) | 35.3 (1.1) | 27.7 (0.8) | 37.2 (1.0) | 30.7 (0.8)* |
| Third | 36.4 (0.8) | 30.1 (0.6) | 35.9 (1.2) | 29.7 (0.9) | 36.9 (1.0) | 30.6 (0.7) |
| Fourth | 37.7 (0.8) | 30.8 (0.6) | 38.4 (1.3) | 29.8 (1.0) | 37.1 (0.9) | 31.5 (0.7) |
| Wealthiest quintile (5[th]) | 37.1 (0.7) | 31.6 (0.6) | 39.9 (1.6) | 31.5 (1.2) | 35.7 (0.8)† | 31.3 (0.7) |

Data are weighted to the 2010 Indonesia population census.

[a] Age-adjusted to the 2010 Indonesian population census.

* $p<0.05$

† $p<0.01$

‡ $p<0.001$ for the difference between urban and rural areas.

primary education was associated with greater prevalence of hypertension in both rural and urban areas. The prevalence of hypertension among the respondents in the lowest wealth quintile was higher in urban areas than in rural areas. However, these significant differences were absent in other wealth quintiles.

Fig 1 illustrates the prevalence of hypertension by wealth quintile and sex in urban and rural areas. It shows that the prevalence of hypertension did not vary significantly by sex or wealth quintiles in either rural or urban areas except in the lowest wealth quintile. The prevalence of hypertension among females in the lowest wealth quintile was higher than among males in that same wealth quintile ($p<0.05$). The hypertension control rates of respondents in the poorest quintile were lower than in any other wealth quintile.

## Awareness, treatment and control of hypertension

Table 3 presents the levels of awareness, treatment, and blood pressure control among respondents with hypertension in the Indonesian population. Overall, 41.8% of respondents (age-adjusted) with hypertension had at some time been diagnosed by a health professional as having hypertension. Respondents in urban areas were more likely (47.4%) to have been diagnosed than those in rural areas (43.7%, $p<0.001$). However, the significance of that difference diminished when the data were adjusted according to the 2010 Indonesia population census. Of those who had hypertension, only 6.6% had taken medication in the previous week. Urban residents were more likely (7.3%) to be receiving treatment than rural residents (5.9%). The overall rate of hypertension control among all hypertensive respondents was only 2.3%, while that among the respondents taking antihypertensive medication was 35.2%.

## Associations between participation in CBHIs for NCDs and hypertension awareness, treatment and control

Fig 2 and S1–S3 Tables present the results of the logistic regressions examining the associations between participation in CBHIs for NCDs, awareness, treatment and hypertension control among all respondents with hypertension, as well as hypertension control among respondents on antihypertensive in all residential areas (Fig 2A), urban areas (Fig 2B) and rural areas Fig 2C). Fig 2A shows that participation in CBHIs for NCDs was associated with higher awareness (odds ratio (OR) = 1.50; 95% confidence intervals (CI) = 1.19; 1.89) and treatment (OR = 2.18; 95% CI = 1.61; 2.95), but not with hypertension control, among all respondents with hypertension (OR = 1.18; 95% CI = 0.84; 1.68) and among respondents on antihypertensives (OR = 1.80, 95% CI = 0.92; 3.52). Among urban-dwelling respondents with hypertension, participation in CBHIs for NCDs was correlated with 38% higher odds of being aware of their disease and 114% higher odds of being on treatment, while respondents with hypertension in rural areas who participated in CBHIs for NCDs had 66% higher odds of being aware of their disease and 114% higher odds of being on treatment.

Middle-aged and older respondents with hypertension had higher levels of hypertension awareness, were more likely to be on treatment, and were more likely to have their hypertension under control than young adults. Similarly, being female, wealthier, having more formal education, and having health insurance showed positive and significant associations with the awareness, treatment and control for hypertension. Respondents with hypertension who were married were more likely to be aware of their disease and to have their hypertension under control.

## Sensitivity analyses

Sensitivity analyses were conducted to check for the occurrence of any misclassification, mainly due to the fact that the outcome (prevalence of hypertension) is common. To address

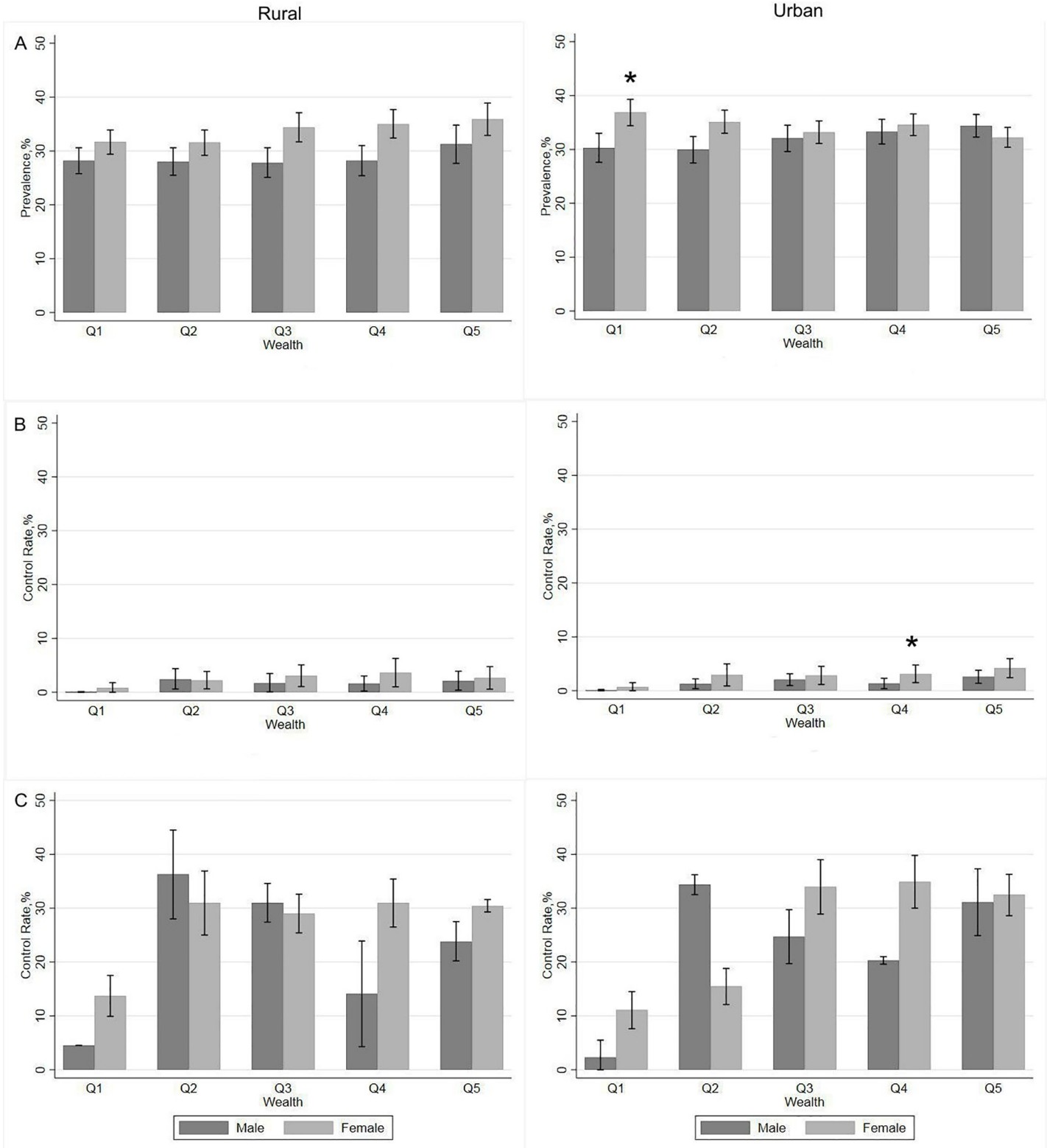

**Fig 1.** Hypertension prevalence and control rates by wealth quintile and sex in urban and rural areas: 1A, Hypertension prevalence; 1B, Control rates in all hypertensive respondents; 1C, Control rates in treated hypertensive respondents. Data are weighted using survey weight and adjusted to the 2010 Indonesian population census. *$p < 0.05$ indicates the differences between males and females. Error bars indicate 95% confidence intervals.

**Table 3.  Awareness, treatment, and hypertension control among respondents with hypertension in the Indonesian population, 2014–2015.**

|  | Total | | Rural | | Urban | |
|---|---|---|---|---|---|---|
|  | **Unadjusted** | **Age-adjusted [a]** | **Unadjusted** | **Age-adjusted [a]** | **Unadjusted** | **Age-adjusted [a]** |
| Awareness | 43.7 (0.5) | 41.8 (0.7) | 43.7 (0.9) | 41.1 (1.1) | 47.4 (0.7)‡ | 42.4 (0.8) |
| Treatment | 10.0 (0.3) | 6.6 (0.3) | 8.4 (0.5) | 5.9 (0.4) | 12.4 (0.5)‡ | 7.3 (0.3)† |
| Control |  |  |  |  |  |  |
| Among all with hypertension | 2.3 (0.2) | 2.3 (0.2) | 2.1 (0.3) | 2.1 (0.3) | 2.4 (0.2) | 2.4 (0.2) |
| Among those treated | 18.6 (0.1) | 35.2 (0.2) | 16.0 (0.2) | 31.3 (0.2) | 20.3 (0.1) | 37.1 (0.2) |

Data are weighted to the 2010 Indonesia population census.

[a] Age-adjusted to the 2010 Indonesian population census.

\*$p<0.05$

†$p<0.01$

‡$p<0.001$ for the difference between urban and rural areas.

this issue, we used Poisson regressions. As can be seen from S4–S6 Tables, the results of the Poisson regressions were reasonably similar to the results of logistic regressions based on a two-sided *p*-value of <0.05.

## Discussion

In this study, we examined the association between participation in CBHIs for NCDs and the awareness, treatment and control of hypertension in Indonesia. To achieve the aim, descriptive analyses were used to describe hypertension rates as well as awareness, treatment and control as the outcome variables of this study. Then, logistic regressions were applied to examine the association between participation in CBHIs for NCDs and the awareness, treatment and control for hypertension among 30,351 respondents aged 18 years and older representing 83% of Indonesia's population in the 2014 IFLS. Our descriptive analyses show that the age-adjusted prevalence of hypertension in Indonesia in 2014 was 30.2%, which was similar to that reported in a previous study in Indonesia [10]. Using the most recent prior wave of IFLS (from 2007), Christiani et al. showed the prevalence of hypertension among urban women aged 15 years and older to be 31% [9]. However, the age-specific rates of hypertension in respondents ages 40 and older in our study were higher (50.2%) than those in a prior study by Hussain et al., whose sample consisted entirely of IFLS 2007 respondents aged 40 years and older (47.8%) [8]. With respect to other studies using samples in the same age range (18 years and older), the age-adjusted prevalence of hypertension that we found in Indonesia was lower than that of Malaysia (35.3%) [52]; it was higher than those of India (25.3%) [53] and the US (29%) [54]. The increasing rate of hypertension among Indonesian adults aged 40 years and above supports studies considering worldwide trends in blood pressure and the prevalence of hypertension [6,7]. Hypertension has been persistently prevalent in LMICs.

The results of our descriptive analyses also show that 41.8% of respondents with hypertension were aware of their condition. This proportion is higher than in the results of prior studies using IFLS 2007 data, in which only one-third of people with hypertension [10] and elevated cardiovascular risk were aware of their conditions. However, awareness of hypertension was higher in urban areas (42.4%) than in rural areas (41.1%). Similar patterns are evident in hypertension treatment. Only 5.9% of rural-dwelling respondents with hypertension were on treatment for the condition. That proportion was slightly higher in urban areas (7.3%). These findings suggest that inequalities between urban and rural areas persist where hypertension care is concerned, although the government has made efforts to increase health care access,

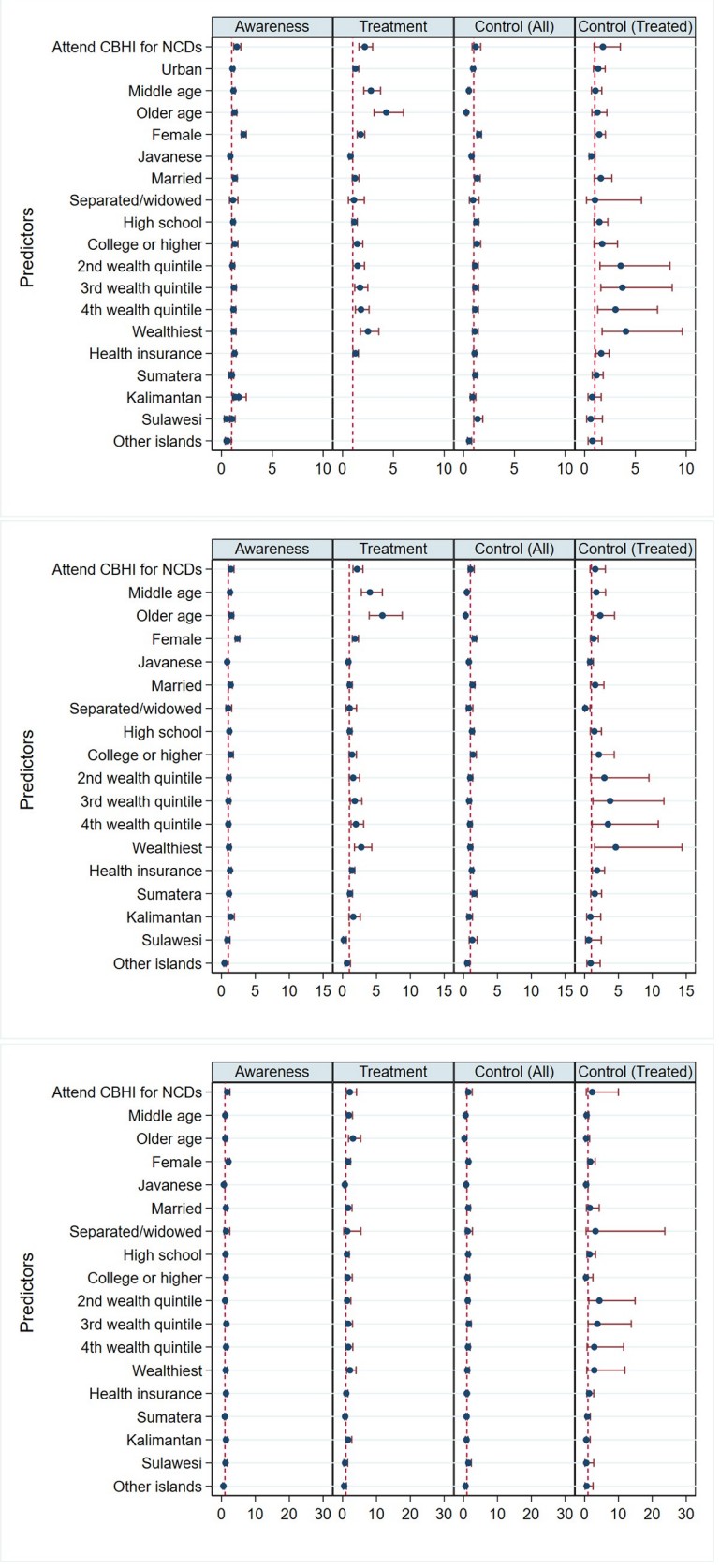

**Fig 2.** Participation in community-based health interventions (CBHIs) for non-communicable diseases (NCDs) and other determinants of awareness, treatment, and control among all respondents with hypertension as well as control among treated respondents in: 2A, all areas; 2B, urban areas; and 2C, rural areas. Data are weighted using survey weight. Reported are odds ratios (95% confidence intervals).

including providing universal health coverage through the launch of *Jaminan Kesehatan Nasional* (JKN) in the early 2014.

The main findings from logistic regression show that respondents with hypertension who participated in CBHIs for NCDs had 37% and 93% higher odds of being aware of and receiving treatment for the condition. This finding supports a review of several cardiovascular risk and hypertension-related community programs which revealed that those programs have been successful in identifying and educating people at risk [55]. In the Indonesian context, the main activities of *Posbindu PTM* and *Posyandu Lansia* focus on health education and providing screening for NCD risk factors, mainly blood pressure, blood glucose and cholesterol [32]. Accordingly, these activities are conducted by trained volunteers from the community (*Kaders*) who share the ethnicity, religion, language, socio-economic status, and life experiences of the people they serve [32]. Culture greatly affects health understanding, and as members of the community, *Kaders* are trusted and form remarkably close bonds with members of the community. Through these culturally sensitive health education strategies, CBHIs in Indonesia may increase awareness of hypertension and encourage those with the condition to receive treatment. The effectiveness of such strategies compared with those carried out by traditional physicians and nurses has also been reported in previous studies [24]. Moreover, Wu et al. found that individuals with hypertension who attended community based health promotion programs demonstrated higher levels of disease management self-efficacy [56]. Including interventions targeting common risk factors such as unhealthy diet and physical inactivity in CBHIs for NCDs in Indonesia may lower the burden of hypertension and that of other chronic diseases in the community.

However, we found that participating in CBHIs for NCDs had no significant relationship with controlled blood pressure among respondents. These findings contrast with prior studies in high-income and LMICs countries that show CBHIs to benefit participants' blood pressure control [57–59]. For example, Bunting et al. and Gonzales et al. showed that individuals who participated in long-term community-based programs for hypertension control were more likely to have their blood pressure under control due to increased access to essential medication and services at nearby locations [59,60]. Hence, one possible explanation for these contrasting findings may be that CBHIs in Indonesia are currently unable to deliver ample essential medication for hypertension. In practice, only a few CBHIs are able to supply sufficient amounts of essential medication for their participants. A lack of availability of essential medication for NCDs in primary public health care has also been widely reported across LMICs [61,62].

Another potential explanation for the contrast in findings relates to the low proportion of adults who participate in Indonesian CBHIs. We found a total of only 4%, with a mere 3% in rural areas. In a previous study, several reasons for low participation among Indonesians were cited, including the feeling that going to CBHIs for NCDs was not important, lack of support (particularly from family), and physical impairment [63]. It is thus important to continue conveying knowledge about the role of CBHIs for NCDs in society as well as about the importance of providing sufficient essential medication for cardiovascular diseases in primary public health care across the country.

Several limitations of this study should be acknowledged. The first limitation is related to the absence of information regarding the duration, frequency and mode of respondents'

participation in CBHIs. Moreover, only 4% of respondents had participated in a CBHI in the 12 months prior to interview. In addition, this study did not capture non-pharmacological treatment strategies such as modifications in diet and physical activity, which may have yielded overestimation of the effect of CBHI participation on the reduction of hypertension rates. This study also used self-reported information on the awareness and treatment of hypertension, which may have led to potential bias due to recall issues or subjectivity in reporting [64]. Finally, this research was based on complete case analysis only, which entails the potential of attrition bias [65]. However, we used survey weight to minimise the possibility of this bias.

Despite these limitations, the present study makes several noteworthy contributions to the literature on CBHIs and NCD prevention as well as assisting policy makers in determining the role of CBHIs in increasing hypertension awareness and treatment in Indonesia. First, our findings have established evidence of the benefits of CBHIs for health care and health. While most previous studies have reported the benefits of CBHIs for communicable disease prevention, access to nutrition and family planning [18,19,66], we show evidence of a link between participation in CBHIs for NCDs and levels of awareness, treatment and control of hypertension. Second, participation in CBHIs for NCDs is associated with awareness and treatment among respondents with hypertension, but not with controlled blood pressure. For policy makers, these findings suggest the need for additional strategies and services for hypertension prevention and control, including providing health education programs and other preventive approaches in the management of hypertension.

## Supporting information

**S1 Table. Logistic regression results of participation in community-based health interventions (CBHIs) for non-communicable diseases (NCDs) and other determinants of awareness, treatment, and control among respondents with hypertension as well as control among treated respondents in Indonesia.**
(DOCX)

**S2 Table. Logistic regression results of participation in community-based health interventions (CBHIs) for non-communicable diseases (NCDs) and other determinants of awareness, treatment, and control among respondents with hypertension as well as control among treated respondents in urban Indonesia.**
(DOCX)

**S3 Table. Logistic regression results of participation in community-based health interventions (CBHIs) for non-communicable diseases (NCDs) and other determinants of awareness, treatment, and control among respondents with hypertension as well as control among treated respondents in rural Indonesia.**
(DOCX)

**S4 Table. Poisson regression results of participation in community-based health interventions (CBHIs) for non-communicable diseases (NCDs) and other determinants of awareness, treatment, and control among respondents with hypertension as well as control among treated respondents in Indonesia.**
(DOCX)

**S5 Table. Poisson regression results of participation in community-based health interventions (CBHIs) for non-communicable diseases (NCDs) and other determinants of awareness, treatment, and control among respondents with hypertension as well as control**

among treated respondents in urban Indonesia.
(DOCX)

**S6 Table. Poisson regression results of participation in community-based health interventions (CBHIs) for non-communicable diseases (NCDs) and other determinants of awareness, treatment, and control among respondents with hypertension as well as control among treated respondents in rural Indonesia.**
(DOCX)

## Author Contributions

**Conceptualization:** Sujarwoto Sujarwoto, Asri Maharani.

**Data curation:** Sujarwoto Sujarwoto, Asri Maharani.

**Formal analysis:** Sujarwoto Sujarwoto, Asri Maharani.

**Funding acquisition:** Sujarwoto Sujarwoto.

**Investigation:** Sujarwoto Sujarwoto.

**Methodology:** Sujarwoto Sujarwoto, Asri Maharani.

**Project administration:** Sujarwoto Sujarwoto.

**Resources:** Sujarwoto Sujarwoto.

**Software:** Sujarwoto Sujarwoto.

**Supervision:** Asri Maharani.

**Validation:** Sujarwoto Sujarwoto, Asri Maharani.

**Visualization:** Sujarwoto Sujarwoto, Asri Maharani.

**Writing – original draft:** Sujarwoto Sujarwoto, Asri Maharani.

**Writing – review & editing:** Sujarwoto Sujarwoto, Asri Maharani.

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
