## [Decision Letter · Decision Letter 0]

18 Sep 2020

PONE-D-20-22592

Community-based health care interventions and the cascade of care for hypertension in Indonesia

PLOS ONE

Dear Dr Sujarwoto,

Thank you for submitting your manuscript to PLOS ONE. After careful consideration, we feel that it has merit but does not fully meet PLOS ONE’s publication criteria as it currently stands. Therefore, we invite you to submit a revised version of the manuscript that addresses the points raised during the review process.

Reviewers pointed out important comments regarding methods ad analysis of data. Please address the comments accruing from the reviewers and resubmit for consideration

We look forward to receiving your revised manuscript.

Kind regards,

Geofrey Musinguzi, MPH, PhD

Academic Editor

PLOS ONE

Journal Requirements:

Reviewers' comments:

Reviewer's Responses to Questions

**Comments to the Author**

1. Is the manuscript technically sound, and do the data support the conclusions?

Reviewer #1: Yes

Reviewer #2: Partly

Reviewer #3: Yes

2. Has the statistical analysis been performed appropriately and rigorously? 

Reviewer #1: Yes

Reviewer #2: No

Reviewer #3: Yes

3. Have the authors made all data underlying the findings in their manuscript fully available?

Reviewer #1: Yes

Reviewer #2: Yes

Reviewer #3: Yes

4. Is the manuscript presented in an intelligible fashion and written in standard English?

Reviewer #1: Yes

Reviewer #2: Yes

Reviewer #3: Yes

5. Review Comments to the Author

Reviewer #1: Thank you for inviting me to review this article. By reporting on population-wide impacts of CBHI programs, where past research has report pilot or initial RCT findings, the current manuscript makes an important and novel contribution to the literature on CBHI for non-communicable diseases in low- and medium-income countries. The manuscript is well written and, aside from a few issues the methodological/statistical, the analysis is robust and well reported.

However, I do have some concerns with the article which I feel must be addressed before I feel that it is suitable for publication. These can largely be split into three categories: [1] issues with the methodology, particularly concerning the time-period over which the data was collected, [2] statistical issues, particularly the lack of control for multiple comparison, [3] structural and content issues with the text, particularly in the discussion.

I have split by comments into major and minor comments which are bullet pointed below. I feel that all these issues can be addressed by the author in some way and once done I believe that the manuscript could be suitable for publication.

Major Comments:

• I was a little confused about the time periods that the CBHI program was launched in relation to the data collection period for the IFLS-5. The author states that the CBHI programs were launched in 2015 however the IFLS-5 was collected in 2014. Based on this reading the data will not capture the relevant time-period to analyse the policy. Perhaps there is a history of CBHI which predates the 2015 policy intervention however one might expect that the implementation of the 2015 policy will have had impacts over the last 5 years meaning that the findings of this paper are somewhat out of date. It may just be that the author needs to add more clarity to the setting section of the introduction to explain this issue to the reader.

• I believe that the p-threshold used in the analysis is too lenient. Given that the author has conducted multiple comparisons of a large sample size I believe corrections for multiple comparisons should be applied. This is mediated by the face that CIs are reported however I think if the analysis is going to rely on CIs alone then 99% C/is should also be reported.

• I wondered whether ethnicity and geographic region could be included as IVs for the logistic regression given that Indonesia is a culturally diverse and geographically spread out country. The author picks up on the latter of these issues by reporting that “CBHIs are inequitably distributed across Indonesia” (Ln 134).

• I think the discussion section needs some work. It begins with a summarised repetition of the introduction which I believe is not necessary before addressing three key findings, two of which are not novel and are not related to aims and objectives of this study. I think the discussion should focus more on the CBHI and should open with a section which covers how the analysis addressed the aims and objectives of the study.

• The figures are in a low resolution and the text is often too small to read, particularly for figure 1.

Minor comments:

• I had some concern about the way the author defined control of hypertension (Ln 168-170) as it does not seem to account for those individuals who may have controlled their hypertension through behaviour change or other treatments. Perhaps this comment is not relevant because CBHI programs in Indonesia may focus specifically on antihypertensive medication. I am not sure, and I am happy to be corrected on this issue as I am not an expert on Indonesian CVD healthcare policy. However, the lack of coverage of behavioural adjustments in article’s definition of control may account in part for the lack of association between CBHI and hypertension control in the analysis where other research has found such associations.

• I think there needs to be more justification of the decision to split the analysis by residential area. Clearly, it is an important distinction when it comes to this issue however I feel as though the author has not described why such a decision was made going into the analysis. Currently, this decision feels like it comes out of nowhere on Ln 186. Additionally, it was not clear how rural vs urban was defined.

• Percentage rates for attendance of CBHIs is reported in the discussion but not in the results. I believe it should be reported in the results section.

• Sentences on Ln 82-84 beginning with “individuals…” should be reworded as they are a little unclear, particular the second sentence beginning with “Otherwise…”

• I don’t believe that the references for the “low-manpower claim” justify this statement specifically for Indonesia (Ln83-84).

• In the abstract and in the methods section (Ln 155) the author refers to participation rates of 93%. Can the author report what this refers to? 93% of whom. Does it mean that of those contacted to take part, 93% agree, or that the retention rate from the IFLS-4 was 93%. I couldn’t find the data in the cited RAND Corp article.

• Could the author specify which survey weight was used in this study as I expect the survey will have multiple weights. If so, why was that survey weight chosen specifically? (Ln 186)

• On lines 283-286 the author refers to two correlation analyses which area not sufficiently reported on with CIs and R-values.

Reviewer #2: 1. Is the manuscript technically sound, and do the data support the conclusions?

The manuscript is technically sound. However, the authors should clearly state the community-based health care interventions they are referring to. There is lack of consistence in use of the words in the objective. Is it (factors associated or risk factors or determinants or predictors or relationship) with awareness, treatment and control of hypertension? This should be made consistent in the entire manuscript bearing in mind the study design.

"NCD" was already abbreviated in the abstract, hence no need to repeat it in the introduction.

What were the health volunteers trained in? Who trained them and did they follow a curriculum? For how long are they trained? What do the authors mean by modest funding from ministry of health and what does it support? "IFLS" was also already abbreviated and there is no need to repeat it.

The lower age cut off for the IFLS-5 participants is not clear (<1).

Let the authors give details of the data that was used. In which wave did these individuals come from? When was this data collected? Did the analysis use the longitudinal data or one survey? What is the study design?

The authors should give more details on the Omron self-inflating sphygmomanometers. Who is the manufacturer, in which country, city and when were they last calibrated?

The authors should indicate clearly the questions that were included in the survey and were relevant to their study. Was it easy for participants to know if their BP value?

What power did the study have given the sample size of 30,351 respondents that was used? How were the control variables selected? What was the justification for categorizing each of the variables included in the study?

2.Has the statistical analysis been performed appropriately and rigorously?

The authors do not show the rigor in performing the analysis in systematically and in detail. What do they mean by "Survey weight was used in all analyses to adjust for non-response bias"? Details of what is meant and how it was done should be included.

The word standard deviation (SD) should be included after the word "mean".

The outcome (prevalence of hypertension) is common. Was use of OR the best option? Use of OR when the outcome is common over estimates the measure of association (Barros, A.J. and Hirakata, V.N., 2003. Alternatives for logistic regression in cross-sectional studies: an empirical comparison of models that directly estimate the prevalence ratio. BMC medical research methodology, 3(1), p.21)

3. Have the authors made all data underlying the findings in their manuscript fully available?

Yes

4. Is the manuscript presented in an intelligible fashion and written in standard English?

Yes

Reviewer #3: Dear Editor,

Thank you for the opportunity to review this manuscript. Congratulations to the authors for the good piece of work that is largely well written. The study aimed at examining the association of community-based health care interventions with hypertension awareness, treatment and control among Indonesians. The work is based on analysis of a large representative dataset and the employed sound techniques. My major criticism is the presentation and portrayal of this work as one from a longitudinal study, yet the data was from a cross-sectional survey in that the results, discussions and discussions seem overstated. Please see below for my specific comments.

Title

The title is misleading and gives an impression of the manuscript being based on a longitudinal evaluation of the community-based health care intervention which was not the case and as cross-sectional data were analysed instead.

Suggested title: Participation in a CBHI and its association with hypertension awareness, control and treatment in Indonesia.

Introduction

In addition to stating the gap as whether CBHIs improve awareness, treatment and control, can we highlight the gaps around the association between the variables as a starting point as we not technically studying the intervention effect.

Line 100: indicate that this is in Indonesia though it still seems overstated.

The aim is correctly stated as examining the association between participation in CBHI and hypertension awareness, treatment and control.

Line 103 to 110 should be omitted. It doesn’t add much information to the study. Only the study aim should be strengthened.

Material and methods

Can you move setting to material and methods?

Line 130 should be a continuation with previous paragraph and not stand on its own.

Line 166 ‘on treatment’ not ‘in’

Line 183 How was the question for health insurance phrased or created?

Provide more details about data collection. What tools were used? was it pretested, validated? how was data collected and managed?

Results

Line 203 & 204 an omit ‘blood pressure’ after systolic.

Line 207 not clear what is meant by ‘lower’ here.

Line 209 was the proportion 3% or 4%? In line 351, you indicate 4%. Cross check.

Line 214 Can we say respondents instead of sample

Table 1 As indicated for means as SE, can we include the same for proportions and include SE. For participation in CBHIs, include the time period of ‘previous 12 months’

Line 227 omit ‘a’ primary

Line 275 Replace with meaningful title based on what the presented data is about

Line 276 to 294 use the past tense in describing the results and present results as those from cross sectional data. For example, instead of saying that participation in CBHI is associated with higher awareness, state that having participated in CBHI was associated with …. Or those who had participated had greater odds for awareness, .….. you can also include the 12 months duration since it is not lengthy and helps to contextualise participation. These edits would need to be extended to the discussion section when results are described.

Line 282 replace ‘and’ with ‘or’ and instead of treated respondents, can we have ‘respondents on antihypertensives’?

Line 283 to 286 where is the 99% and 50% coming from? Where is it referenced? If data is not shown for some results, state so

Discussion

Line 297 to 307 Omit paragraph as it is a repetition of the introduction information and doesn’t add much to your study.

Line 308 to 320 As you make comparisons of hypertension prevalence, can you start with comparisons in Indonesia (312 to 320) before we bring in other countries?

Line 334 to 349 All comparisons made are based on longitudinal studies which is misleading. It is thus not proper to mention that this study contradicts prior longitudinal studies, yet this was across sectional study and so the comparisons are not similar. It would be better to make appropriate comparisons with cross sectional studies or contextualise the longitudinal ones.

Line 347 Could use ‘may’ instead of ‘will’

Line 351 in addition should be a continuation and not a new paragraph.

Line 354 Where? In Indonesia?

Line 361 to 364 talks about the fact that many respondents did not have controlled BP but this is not a study limitation.

Line 364 to 365 Absence of medication data doesn’t seem to fit our study objectives and not carrying out further investigations does not affect our study to state it as a limitation.

Line 365 How would not capturing non-pharmacological treatment strategies lead to an underestimation of treatment rates? It could have led to an overestimation of the CBHI effect which limitation should be highlighted.

The fact that the participation of respondents could not been quantified is also a study limitation and thus it is not clear whether there was a relationship with duration, frequency or mode of participation with the outcomes. Moreover, the participation rate was very low (4%).

Line 373 The stated contributions are overstated, and some are not in context with the study findings. Revise these

Abstract

Conclusion is misleading. The CBHI described does not include lifestyle modification on which the conclusion is based.

General

The writing is laden with repetitions, inconsistencies in grammar and is not succinct. A through grammatical check will help improve it. For example repetitions around ‘the respondents’ after it has already been mentioned.

References

Reference 3 Edit world Health organization as author.

6. PLOS authors have the option to publish the peer review history of their article (what does this mean?). If published, this will include your full peer review and any attached files.

Reviewer #1: **Yes: **Thomas Grice-Jackson

Reviewer #2: No

Reviewer #3: No

---

## [Author Response · Author response to Decision Letter 0]

23 Oct 2020

We submitted all responses to each reviewer in the attached files.

---

## [Decision Letter · Decision Letter 1]

8 Dec 2020

Participation in community-based health care interventions (CBHIs) and its association with hypertension awareness, control and treatment in Indonesia

PONE-D-20-22592R1

Dear Dr. Sujarwoto Sujarwoto,

We’re pleased to inform you that your manuscript has been judged scientifically suitable for publication and will be formally accepted for publication once it meets all outstanding technical requirements.

Kind regards,

Geofrey Musinguzi, MPH, PhD

Academic Editor

PLOS ONE

Additional Editor Comments (optional):

Based on the collection of reviews and responses to the reviews, feedback from reviewer three and my own review, I find the comments satisfactorily addressed.

Reviewers' comments:

Reviewer's Responses to Questions

**Comments to the Author**

1. If the authors have adequately addressed your comments raised in a previous round of review and you feel that this manuscript is now acceptable for publication, you may indicate that here to bypass the “Comments to the Author” section, enter your conflict of interest statement in the “Confidential to Editor” section, and submit your "Accept" recommendation.

Reviewer #3: All comments have been addressed

2. Is the manuscript technically sound, and do the data support the conclusions?

Reviewer #3: Yes

3. Has the statistical analysis been performed appropriately and rigorously? 

Reviewer #3: Yes

4. Have the authors made all data underlying the findings in their manuscript fully available?

Reviewer #3: Yes

5. Is the manuscript presented in an intelligible fashion and written in standard English?

Reviewer #3: Yes

6. Review Comments to the Author

Reviewer #3: (No Response)

7. PLOS authors have the option to publish the peer review history of their article (what does this mean?). If published, this will include your full peer review and any attached files.

Reviewer #3: No

---

## [Editor Report · Acceptance letter]

15 Dec 2020

PONE-D-20-22592R1 

Participation in community-based health care interventions (CBHIs) and its association with hypertension awareness, control and treatment in Indonesia 

Dear Dr. Sujarwoto:

I'm pleased to inform you that your manuscript has been deemed suitable for publication in PLOS ONE. Congratulations! Your manuscript is now with our production department. 

Kind regards, 

on behalf of

Dr. Geofrey Musinguzi 

Academic Editor

PLOS ONE